# Calibration and Validation of an Autonomous, Novel, Low-Cost, Dynamic Flux Chamber for Measuring Landfill Methane Emissions

**DOI:** 10.3390/s25216613

**Published:** 2025-10-28

**Authors:** Avery G. Brown, Nikona G. Rousseau, Dylan Doskocil, Cullen T. O’Neill, Seth G. VanMatre, Justin J. Kane, Joanna G. Casey, Michael P. Hannigan, Evan R. Coffey

**Affiliations:** 1Department of Physics and Engineering, Fort Lewis College, Durango, CO 81303, USA; 2Department of Mechanical Engineering, University of Colorado at Boulder, Boulder, CO 80309, USA

**Keywords:** methane, dynamic flux chamber, low-cost, metal oxide, landfill emissions

## Abstract

**Highlights:**

**What are the main findings?**

**What is the implication of the main finding?**

**Abstract:**

A low-cost, dynamic flux chamber optimized for landfill emissions measurement was designed, fabricated, calibrated, and validated for measurements of methane flux ranging from 0 to 150 g/m^2^-day. A centrifugal blower fan and a flow meter were plumbed in series to draw a bypass flow through the flux chamber. Both ambient and chamber methane concentrations were measured using the arrays of four low-cost metal oxide sensors. Leveraging the sensors’ overlapping sensitivity to changes in methane concentration, temperature, and humidity, multiple linear regressions were trained on laboratory data and combined into a piecewise methane calibration function. An algorithm was developed to select the most useful interaction terms among all sensor responses to optimize the predictors in each model. The piecewise regions for methane measurement were 0–100 ppm, 100–1500 ppm, and 1500–12,000 ppm. The root mean squared errors for each piecewise region were 3.1 ppm, 21 ppm, and 307 ppm, respectively. Controlled quantities of methane were delivered to the flux chamber in a laboratory setting for validation. Measurements yielded good agreement with an RMSE and MBE of 7.3 g m^−2^ d^−1^ and 2.2 g m^−2^ d^−1^, respectively. The flux chamber was tested at a closed landfill to validate its ability to autonomously and continuously operate in the field.

## 1. Introduction

Swift and decisive action is required to reduce greenhouse gas emissions if we are to lessen the severity of negative repercussions that stem from climate change [1]. Methane has a high global warming potential and warms the Earth 80 times more effectively than an equivalent mass of carbon dioxide, over a 20-year time horizon [2]. Anthropogenic emissions of methane range from fossil fuel production, livestock farming, agriculture, and landfills. Landfills account for 17% of US anthropogenic methane emissions and stem from the breakdown of organic material [3]. Measures have been put into place to capture methane from landfills, and either utilize it as a fuel or burn it, to minimize its environmental impact. Not all landfills possess this technology, and even landfills that do still emit significant amounts of methane. Determining the amount of methane being emitted from a landfill, and where the methane is emitted from is often difficult [4]. Additional measurements are needed to quantify methane flux, which will improve our understanding of where methane is emitted from on a landfill and the total amount emitted. This knowledge can then inform how these emissions can be captured and mitigated.

### 1.1. Techniques to Measure Emissions from Landfills

Measurement techniques that are used to estimate methane emissions from landfills encompass a broad range and include, among others, arial surveys using both remote sensing instrumentation [5] and in situ methane measurements coupled with a mass balance modeling [6,7], eddy covariance [8], fence-line surveys coupled with tracer gas and gaussian plume modeling [9], models informed by atmospheric concentrations measured by sensor networks [10], flux chamber measurements [11,12,13], and more recently, targeted satellite observations [14]. All these methods rely on some form of modeling coupled with the measurement tool. Some of these methods, such as eddy covariance towers and satellite imaging, lack the ability to resolve the spatial distribution of emissions across the landfill topology [15]. Flux chambers can be used to measure methane emissions from specific locations. While flux chambers can only measure emissions from a small fraction of the total landfill area at a time, they represent one of the most direct measurement techniques for methane emissions from the ground. Flux chambers can be configured as static, with a closed loop measurement of the methane concentration in the chamber, or dynamic, with a bypass flow of ambient air through the chamber and measurements of methane in both the ambient air and the chamber [16]. Static flux chambers work by measuring the concentration of methane as it accumulates over time [16].

Although static flux chambers are simple in design, they can be inherently inaccurate because methane flux is artificially reduced when chamber concentrations rise far above ambient levels [17]. An autonomous, and continuously operating, static flux chamber would need to be periodically opened and flushed. Rather than accumulating methane like the static chamber, the dynamic chamber allows methane flux from the ground to flow through the system where it is measured and then exhausted into the atmosphere [16]. Dynamic flux chambers are often more complex than static chambers but limit the possibility of reducing or inducing flux from the ground and do not require a periodic flushing mechanism when deployed for continuous operation [16]. A series of dynamic flux chamber measurements at deployment locations across a landfill can yield reliable methane emission estimates [18]. The strengths and weaknesses of the tools and technologies used to measure landfill emissions mentioned above are summarized in Table 1. This summary of methane quantification techniques is intended to contextualize the advantages and limitations of dynamic flux chambers compared with other common emission quantification methods, rather than to serve as a comprehensive review.

### 1.2. Design Considerations for a Dynamic Flux Chamber

When ambient and chamber methane concentrations are known, in addition to the flow rate through the chamber, flux can be calculated using a mass balance. The chamber is modeled as a well-mixed system, where the mass of methane is conserved. Figure 1 shows a schematic of the pathways for methane to enter and exit the chamber, where *F* is the methane flux, *Q* is the flow rate through the dynamic flux chamber, *C_b_* is concentration of methane passing through the flux chamber, *C_a_* is concentration of methane in the surrounding ambient air, *V* is the chamber volume, and *A* is the surface area of ground covered by the chamber.

The associated mass balance equation for the box model is presented in Equation (1).(1)$$\frac{dC_{b}}{dt}V=FA+QC_{a}-QC_{b},$$

Methane flux can be calculated as a function of the following measurements: flow rate, ambient methane concentration, chamber methane concentration, and the rate of change in the chamber methane concentration with time. Equation (1) can be rearranged into Equation (2) to solve for flux.(2)$$F=\frac{\frac{dC_{b}}{dt}V+Q(C_{b}-C_{a})}{A}$$

If the methane concentration within the chamber reaches steady state, we can simplify Equation (2), setting $\frac{dC_{b}}{dt}$ to zero, to solve for methane flux using Equation (3).(3)$$F=\frac{Q(C_{b}-C_{a})}{A}$$

While flux chambers provide relatively direct emissions information with good spatial specificity, they are very limited in their spatial breadth of measurement for a given moment in time. Given the complex nature of methane emissions from large area sources, like landfills, both strong spatial and temporal variability are expected. Developing methane flux chambers with a low price point would allow for the deployment of a fleet of many flux chambers capable of continuous operation to better characterize both spatial and temporal variations in emissions. Since costly methane sensing mechanisms drive the price point of currently available flux chambers, employing low-cost sensors with useful signal to noise ratios for a given flux chamber design and application is key. A continuously operating static methane flux chamber, with periodic flushing when a concentration threshold was reached, was developed using a low-cost optical methane sensor to measure emissions from landfills [19]. This flux chamber was deployed on a landfill in Michigan for nine months and provided insight into seasonal, diurnal, and weather-induced variations in the variability of flux. Bastviken and colleagues also developed a static methane flux chamber using low-cost sensors [20]. To our knowledge, we are developing the first dynamic methane flux chamber that operates using low-cost sensors.

### 1.3. Measuring Methane with Low-Cost Sensors

A dynamic methane flux chamber requires measurement of the methane concentration within the chamber, which is modeled as a well-mixed volume, as well as measurement or knowledge of the methane concentration in the bypass flow that is continuously being drawn into the chamber. Infrared (IR) gas analyzers are commonly used to measure methane within the chamber by measuring the absorption of light at a targeted wavelength (absorption band) when exposed to a sample. Although this detection method is excellent for distinguishing methane from other gases, methane gas analyzers are very expensive and cost upwards of 40,000 USD. Low-cost IR sensors are also emerging on the market for upwards of 100–200 USD, like the sensors used by Draughon and colleagues in their methane flux chamber design [19,21], but they struggle to measure near-atmospheric methane concentrations (<100 ppm), limiting the detection limit of a flux chamber [22,23]. Metal oxide (MOX) sensors, like the Figaro TGS 2600 and TGS 2611 sensors [24,25], are more affordable and more sensitive to methane than low-price-point optical sensors. These sensors have been used to measure methane concentrations, even down to atmospheric concentrations, in a growing number of studies [26,27,28,29,30,31,32,33,34]. These MOX sensors have an affordable price point of around 10–13 USD per sensor. However, MOX sensors are not selective to methane. They respond to variability in other volatile organic compounds (VOCs), as well as variations in humidity and temperature.

Previous studies have shown that arrays of several MOX sensors with overlapping sensitivities, including the TGS 2602 sensor, which is not sensitive to methane directly but has overlapping sensitivity with the TGS 2600 and 2611 to confounding gases, can be utilized together to effectively measure methane concentrations, even in the presence of confounding gases [26]. With specific relevance to the current study, a similar sensor array with this combination of sensors has even been utilized to measure methane concentrations on a landfill recently [35]. For ambient methane measurements, calibration equations for these sensor arrays can be developed by means of field normalization by co-locating these MOX sensor arrays with a calibrated reference instrument in an ambient environment that experiences variable methane concentrations, temperature, and humidity, and then performing a multiple linear regression [26]. This involves using each sensor’s signal, along with temperature and humidity sensor signals, and in some cases the interaction ratios of these signals, as predictors, each with an associated coefficient.

While these low-cost MOX sensors have enough sensitivity to respond to ambient variations in methane, they also have a very wide measurement range and are not expected to experience irreversible chemistry (poisoning) below methane concentrations of 12,500 ppm according to a representative from Figaro. This sensitivity in combination with a wide measurement range makes these sensors an interesting option for measuring methane in a low-cost flux chamber where both sensitivity and a wide range of measurement are of value. Bastviken and colleagues developed a low-cost static methane flux chamber using a Figaro TGS 2611-E13 sensor and explored an alternative to extensive field model training with an expensive reference instrument [20]. This alternative involved a periodic single measurement rather than continuous sampling. Bastviken and colleagues also emphasized the importance of comprehensive lab testing involving all relevant atmospheric conditions and their effect on the methane sensors. This should help account for as many cross-sensitivities as possible in a lab setting such as humidity, temperature, pressure, and the presence of common atmospheric pollutants. In this work we develop a low-cost dynamic flux instrument optimized to measure methane emissions from landfills. We demonstrate the calibration and validation of an array of MOX sensors using a multiple linear regression to quantify methane.

## 2. Materials and Methods

A dynamic flux chamber was designed, optimized, and fabricated in consideration of sample surface area, inlet size, cost effectiveness, power consumption, and portability. Two arrays of four MOX sensor voltage outputs, measuring ambient and flux chamber methane concentrations, were calibrated across a range of temperature and humidity conditions to obtain a calibration applicable to field measurements. Both laboratory and field tests were conducted to validate the performance and accuracy of the instrument. The laboratory experiments were carried out at Fort Lewis College in Durango, CO while the field experiments were performed at a closed landfill in southwest Colorado.

### 2.1. Flux Chamber Design and Optimization

A dynamic flux chamber, shown in Figure 2, was constructed using an aluminum brazier pan with a diameter of 18.5 inches sized to maximize surface area of the sample while still being easily portable. Surface area was maximized to minimize the percent uncertainty in our internal chamber CH4 measurements, since a larger surface area allows for capturing more CH4 at a given flux rate, resulting in higher internal concentrations. Figure 3 summarizes a sensitivity analysis we performed to determine how the diameter of the chamber would influence uncertainty in our internal chamber CH4 measurement. Minimizing the percent uncertainty in internal CH4 concentrations was an important design consideration since this parameter was expected to be the primary driver of our overall uncertainty in measured flux rates. In this analysis we assumed a constant uncertainty of 10 ppm for internal CH4 measurements, a flux rate of 0.1 g/m^2^/day, and flow rate of 2.8 lpm. The 5 inch depth of the brazier pan was selected to accommodate rugged terrain while minimizing volume to minimize time to steady state. The volume of the chamber influences time to steady state within the chamber but does not influence steady state internal CH4 concentrations in the chamber, so height of the chamber was not considered in the sensitivity analysis for the uncertainty of internal CH4 concentrations.

During operation, ambient air is pulled through an inlet (ventilation cap) into the chamber and subsequently an internal sensor enclosure and a flow meter by a BFB0512HHA-C 12V DC blower fan. The size of the inlet was maximized to be as large as possible without allowing air from the chamber to escape to keep the pressure within the chamber as close as possible to atmospheric pressure, since small deviations from atmospheric can greatly influence flux rates [18,36,37,38]. The maximum inlet size was experimentally determined to be one quarter inch using a smoke generator to identify any flow from the chamber escaping through the inlet We measured the difference between the pressure under the flux chamber and the ambient air to be an average of 11.8 Pa during a normal operation using BME680 sensors. Average diurnal variations in pressure in the ambient environment over the course of three weeks were measured with the same set of BME680 sensors to be 198 Pa, and the full-scale range of ambient pressures measured during the same period was 515 Pa. The pressure differences imposed by the chamber were small relative to both average diurnal variations in pressure and full-scale variations in pressure, at 6% and 2.3%, respectively.

Two separate copper sample lines from the flux chamber were included to promote the measurement of a well-mixed concentration within the chamber. Given the importance of a known flow rate through the chamber in the quantification of flux, a Sensirion SFM4300-20-P flow meter was installed after the internal sensor enclosure. The decision to use a blower fan rather than a pump reduced cost as well as power draw and increased operational lifetime. Figure 4 shows the flow path through the dynamic flux chamber.

Figure 5 shows the steady state concentration in the chamber modeled with Equation (3) across a range of flow rates. We determined the minimum flow rate required to stay under both the lower explosive limit for methane of 50,000 ppm and the poisoning limit of the sensors of 12,500 ppm to be 2.8 lpm at the top of our expected measurements range (150 g/m^2^/day).

Given the goal of hourly minimum time resolution for continuous flux measurements on a landfill, we performed a time to steady state analysis. By performing a separation of variables in Equation (1), and integrating, we arrive at Equation (4), which allows us to compute the concentration in the chamber as a function of time at a given flux rate, where *t* is time.(4)$$C_{b}\left(t\right)=\frac{F*A}{Q}\left(1-e^{-\frac{Q}{V}t}\right)+C_{a}$$

Figure 6 shows the time to steady state numerically modeled in the chamber with Equation (4) as a function of flow rates. We see that with a flow rate of 2.8 L per minute, and our maximum anticipated flux rate, we would achieve steady state in less than 30 min.

The flux chamber sensor enclosure, flow meter, and blower fan were all placed inside a 50 Caliber ammo can for weather protection and durability. A 30 Ah ECO-WORTHY LiFePO4 lithium-ion battery and 30-amp charge controller for solar power control were also mounted inside the ammo can. All wiring and plumbing in and out of the ammo can were performed through NPT cable glands to ensure weather protection. An aluminum die-cast box housing the ambient sensor board, cellular module, and antenna were mounted to the side of the ammo can. The sensors housed in this enclosure measured ambient methane concentrations. This enclosure also housed a pc fan to increase air flow along with an antenna to transmit raw data via cellular communication to a database. The dynamic flux chamber was designed to operate autonomously and continuously, transmitting data every 10 s and powered by an off grid photovoltaic system comprising a 120 W 12 V monocrystalline solar panel, 30 A charge controller, and a 12 V 30 AH LiFePO4 battery, all made by ECO-WORTHY.

### 2.2. Printed Circuit Board Design

Considerable attention was devoted to the design and production of the printed circuit board (PCB), focusing on heat dissipation, signal integrity, and data retention. The design was inspired by a prototype board developed by the Hannigan Air Quality Lab [30]. As shown in Figure 7, the PCB is organized into four functional sections across two distinct boards. The main board integrates ambient sensors, a power management system, and the microcontroller—MKR NB 1500, which enables LTE-m/NB-IoT cellular communication for remote data transmission. In the event of connectivity issues, data is locally buffered and stored onboard. The ‘Internal Sensors’ section is designed to break off from the main PCB and is responsible for measuring methane concentrations within the chamber. Communication across these two boards and the various sensors utilize I2C protocol. Although both boards are functionally distinct, they are fabricated on a single PCB to reduce manufacturing costs.

The core sensing components of the system include several MOX sensors (TGS 2600, TGS 2602, TGS 2611) and a BME680 environmental sensor. These components are sensitive to not only methane, but also to variations in temperature, humidity, and the presence of other VOCs, which can alter the sensor output. Table 2 shows the various sensitivities of each sensor and their costs.

### 2.3. Internal Chamber Sensor Calibration

Due to the varying responses to numerous VOCs, temperature, and humidity, four unique MOX sensors, along with temperature and humidity sensors were used together as an array to measure methane, leveraging their overlapping sensitivities.

A calibration system was built to expose the sensors to varying methane concentrations, temperatures, and humidities for the purpose of building a multiple linear regression model capable of distinguishing changes in methane concentration from changes in temperature and humidity. Known methane concentrations were delivered to the sensors using compressed air, Airgas compressed gas cylinders with NIST traceable methane concentrations, and ALICAT mass flow controllers. Heating to above ambient temperatures was controlled using an aluminum chamber outfitted with Peltier thermoelectric devices, heat sinks, and insulation. Cooling was controlled by thermally pasting and mounting an aluminum block to the bottom of the aluminum temperature control chamber. Ice water was then pumped through the aluminum block, cooling through conduction and forced convection. Humidity was controlled by flowing methane dilutions through either water or desiccant, increasing or decreasing the humidity, respectively.

A LI-COR LI-7819 trace gas analyzer was used as a reference instrument to validate low concentrations (0–100 ppm) of methane delivered to the internal sensor array. Once the raw sensor values reached steady state at a delivered concentration, temperature, and humidity, ten minutes of the steady state data were collected. When enough data was collected across the ranges of methane concentration (0–12,000 ppm), temperature (0–40 °C), and humidity (0–80% RH), the raw data were used to develop, or train, a set of multiple linear regression models with MOX, temperature, and humidity sensor signals and ratios as the predictor variables, and the known methane concentration as the response variable.

We considered using machine learning methods for sensor calibration, but the training datasets generated through our laboratory calibration techniques were not large enough to support the training of artificial neural networks. We did some preliminary exploration of polynomial models but found they did not perform as well as linear models when applied to new data. However, they may be worth exploring further in the future to improve model residuals and further reduce uncertainty of measured methane concentrations, particularly if an automated calibration system could be developed to support larger training datasets.

#### 2.3.1. Predictor Selection

Model predictors (features) were selected using the following methods. Each individual MOX sensor signal was included as a predictor based on the overlapping sensitivities to CH4 and confounding VOCs among these sensors. Temperature and humidity sensor signals were included as predictors due to on their known influence on MOX sensor signals. Finally, ratios between some sensor signals were included based on a Monte Carlo-type analysis in which each possible ratio of sensors was used as a predictor (always in addition to each individual sensor signal alone) to generate unique multivariate linear regressions (MVLRs). Including ratios as predictors was determined to be permissible because we observed their linear response to changes in methane concentration. In the Monte Carlo-type analysis, the ratio that produced the MVLR with the lowest RMSE was selected for each iteration. The next iteration of models was generated using the previously identified best ratio(s) in addition to the individual sensor signals as predictors. For example, ratio 1 was used in the model that identified ratio 2. Both ratios 1 and 2 were used to identify ratio 3 and so forth. Six ratios were included in each MVLR. This method was implemented to find not only the best ratios to use, but the best combinations of predictors within the whole model. Coefficients of each predictor were determined by minimizing the sum of the squared differences between the target methane concentration and predicted values.

While models were optimized based on their RMSE, both the mean biased error (MBE) for each model and *p*-values of predictors in each were also assessed. Although the ratios were helpful to the regression, it was important to limit how many ratios were used to not inadvertently build an over-defined regression. The use of predictor ratios within the multiple linear regression models provided added versatility beyond a simple linear model, while remaining more constrained than a higher-order model and less prone to overfitting. All model training was performed using data collected in the laboratory with controlled delivery of CH4 concentration, temperature, and humidity. These lab-trained models were then tested in both laboratory and field settings to assess performance.

#### 2.3.2. Piecewise

We observed that the MOX sensors exhibited distinct linearity behaviors across different methane concentration ranges. To address this, a piecewise approach was applied to the regression to divide the unique linear sections of the methane response. Each section had its own regression developed using training data in and near the identified region of linearity. Each piecewise section of the methane calibration equation was trained on laboratory data.

The piecewise breakpoints were selected based on several factors. The foremost of these was the MOX sensor response to CH4, temperature, and humidity in the laboratory. There were identifiable regions of pseudo-linearity in sensor responses that informed starting points for the piecewise divisions. The limited size of our training dataset was another important factor. We balanced maximizing the coverage and density of training points in each piecewise range with including enough piecewise segments to have reasonable linearity in each model. A single piecewise breakpoint did not perform well so we increased to two breakpoints (three piecewise segments total) and fine-tuned the training and application ranges from there based on model performance. Due to the very large dynamic range of internal CH4 concentrations we needed to measure, we increased the spacing between CH4 setpoints delivered to sensors logarithmically, so higher values were spaced farther apart than lower values. We selected breakpoints for piecewise models in locations that could firmly bookend the linear models with delivered concentrations.

### 2.4. Ambient Sensor Calibration

The ambient MOX sensors did not require a wide range of methane concentration exposure to develop a regression. These sensors were placed outdoors in an environment with naturally varying CH4 concentrations, temperatures, and humidities. The LI-COR sampled from the same location acted as the reference instrument for methane concentrations. The ambient data collected was also regressed using multiple predictors like the internal sensors. Because a tremendous amount of redundant data is present when using the LI-COR, a 70-30 data split (training–validation) was used to develop the ambient regression.

### 2.5. Flux Chamber Validation

Although performance of calibration models can help inform accuracy and precision of the MOX sensors, additional validation tests were conducted to characterize the instrument’s flux-measuring capability, and field operation. The first validation test carried out was a flux validation test. Known quantities of methane were delivered to the dynamic flux chamber, and internal concentrations were measured using the low-cost sensor array and piecewise calibration model. The steady state box model was applied to quantify flux readings on a continuous basis as well as on a discrete basis, using the average methane measured after stabilization for each setpoint. The transient box model was also applied on a continuous basis for comparison.

The second test was conducted at the decommissioned landfill in southwest Colorado. The LI-COR reference instrument was plumbed to sample the same air as the internal sensor array to verify internal methane measurements in a field setting.

#### 2.5.1. Flux Laboratory Validation

To deliver a known methane flux, the bottom of the flux chamber was sealed, and the ALICAT mass flow controllers delivered a known methane concentration to 5 equally distributed sample ports entering the chamber. Methane was delivered from the NIST traceable gas cylinders and compressed air in some cases until the sensors reached steady state for a period of 10 min. The calibration model was applied to quantify methane concentrations. Then, using the transient and steady state box model equations seen in Equations (2) and (3), methane flux was quantified and compared to the delivered flux. Averages were also taken of the methane concentrations once measurements stabilized for each setpoint, and the steady state model was used to determine a single, discrete flux estimate for each setpoint.

#### 2.5.2. Field Validation of Chamber Concentration

Leveraging a closed landfill in the region, two sessions of methane flux data collection were conducted spanning the month of April in 2024. Unfortunately, flux could not be directly validated in the field without the assistance of another flux-measuring instrument, such as an eddy covariance tower, but internal methane concentrations within the chamber could be more readily validated. The LI-COR gas analyzer was plumbed directly into the sample line going from the flux chamber to the internal sensor array, sampling the air from one section of copper tubing, and returning it to the other. The configuration of this experimental setup can be seen in Figure 8.

Since both the LI-COR and the internal sensor array were exposed to the same concentrations of methane, the LI-COR readings were used to validate the sensor concentration readings after applying the piecewise methane calibration model. During the field validation significant variation was observed in chamber concentration, and thus methane flux. The steady state box model was used to continuously quantify flux during the field deployments.

### 2.6. Flux Quantification

When quantifying methane using the piecewise regression, a method to determine the correct piecewise was required. This was achieved by applying intermediate regressions to hone in on the approximate region of the regression that the measurement falls into, then applying the correct regression. Figure 9 shows the process flow that we employed to determine which piecewise to use to quantify methane from raw sensor signals.

## 3. Results and Discussion

### 3.1. Methane Sensor Calibration

Multiple linear regression models were developed to be applied across three piecewise ranges of methane concentrations, 0–100 ppm, 100–1500 ppm, and 1500–12,000 ppm. Each range was trained using data ranging from 0 to 100 ppm, 100–1500 ppm, and 100–12,000 ppm, respectively. Each regression follows the general format seen in Equation (5),(5)$$C_{b}=C_{00}S_{00}+C_{02}S_{02}+C_{11}S_{11}+C_{BME}S_{BME}+C_{T}S_{T}+C_{H}S_{H}+C_{R1}R_{1}+C_{R2}R_{2}\;+C_{R3}R_{3}+C_{R4}R_{4}+C_{R5}R_{5}+C_{R6}R_{6}+B,$$
where *C_b_* is the concentration of methane in the internal sensor chamber. *C*_00_, *C*_02_, *C*_11_, *C_BME_, C_T_,* and *C_H_,* are the coefficients for the TGS2600, TGS2601, TGS2611, BME, temperature, and humidity predictors, respectively. *S*_00_, *S*_02_, *S*_11_, *S_BME_*, *S_T_*, and *S_H_,* are the raw sensor signals for the TGS2600, TGS2601, TGS2611, BME, temperature, and humidity sensors, respectively. *C_R_*_1_, *C_R_*_2_, *C_R_*_3_, *C_R_*_4_, *C_R_*_5_, and *C_R_*_6_, are the coefficients for the six ratios of sensor values *R*_1_, *R*_2_, *R*_3_, *R*_4_, *R*_5_, and *R*_6_, respectively. And *B* is the intercept. Table 3, Table 4 and Table 5 below communicate the regression equations and the RMSE values for the 0–100 ppm, 100–1500 ppm, and 1500–12,000 ppm piecewise functions, respectively.

Figure 10 shows comparisons between the laboratory-delivered methane concentrations, or the reference concentrations, and the concentrations measured by the sensors using the regression from each respective region of the piecewise, determined by the reference concentration. Figure 10A shows the comparisons across the entire methane concentration measurement range (0–12,000 ppm), and Figure 10B–D show the comparisons within the three piecewise ranges 0–100 ppm, 100–1500 ppm, and 1500–12,000 ppm, respectively. An overlapping one-to-one line can also be seen in each of the four figures, representing the ideal case.

Although the regression results indicate a one-to-one linear relationship between the reference concentrations and measured concentrations, many methane concentrations still carry residuals due to variation from extreme temperature and humidity conditions. The methane concentration residuals for each respective figure above can be seen in Figure 11.

The maximum percent error for each regression range is defined as the ratio between the RMSE for that range and the lowest concentration we aim to measure in that range. This metric is presented along with other key model performance criteria for each section of the piecewise regression in Table 6. The maximum percent error is highest for the low-range regression, at 31.0%. The maximum percent error for the mid- and high-range regressions is similar at 21.1 and 20.5, respectively. Currently, variation in temperature and humidity appears to have an outstanding impact on very low concentration measurements. We think it is likely that much of the noise and error currently present in each regression section could be further resolved and improved using an increased number of redundant data points for training. This suggests that future iterations of the low-cost methane flux measurement could likely benefit most from more training data at low methane concentrations. An RMSE of 3.1 ppm for the low-range regression is encouraging given our lower detection limit goal of 0.1 g m^−2^ day^−1^. Resolving 0.1 g m^−2^ day^−1^ would correspond to resolving 10 ppm from 0, so an RMSE of 3.1 ppm would result in a signal to noise ratio of 3.2 at the bottom end of our targeted flux measurement range.

As observed in the methane concentration residuals, the uncertainty at near ambient methane concentrations is too high to make a measurement, but the uncertainty across the entire measurement range is acceptable.

### 3.2. Flux Quantification and Validation

Using the laboratory flux validation experimental setup described previously, flux was quantified using two approaches. For the validation experiment, mass flow controllers were set to constant flux setpoints, and adequate time was allowed for the methane flux to reach steady state. This meant that the steady state form of the mass balance equation could be applied to calculate flux from the calibrated sensor readings. However, as seen in the time series sensor data, concentrations, and thus flux, almost never reached steady state. To account for this, an additional model was built to calculate flux using the transient box model equation. In Figure 12, both models are applied to 8 h of laboratory-controlled methane flux and compared.

During each apparent steady state region of methane flux, the highlighted error regions overlap for the delivered flux and measured flux using both methods. There is good agreement between the delivered flux and the measured flux. Table 7 shows a comparison of average flux measurements using steady state and transient models relative to our delivered flux setpoints, along with discrete flux measurement performance, attained by computing flux based on the average steady state sensor responses at each setpoint.

Note from Table 7 that flux measurements made using the setpoint average discrete method resulted in a lower RMSE, but a slightly higher MBE relative to the steady state model applied to the continuous data during the test. By averaging only the stabilized portion of each setpoint, we removed transient noise, which reduced variability and resulted in tighter residuals, resulting in a smaller RMSE. But since we only sampled after stabilization, it appears we consistently underestimated flux rates relative to the continuous steady state model, resulting in an increased MBE.

Figure 13 shows flux rates calculated at each setpoint using the steady state model, based on the average sensor response after stabilization. A one-to-one comparison is made with known flux rates delivered with mass flow controllers.

In Figure 13 uncertainty in the delivered flux is based on uncertainty in the methane concentration in NIST traceable gas cylinders and an assumption of ambient methane concentrations (feeding the compressed air) of 2 ppm with an uncertainty of 5% or 0.1 ppm, since ambient methane concentrations in the compressed air ranged from 1.9–2.1 ppm when measured with the LI-COR. Uncertainty in the measured flux is based on the manufacture specified uncertainty of the flow sensor, the standard error of the mean of the flow at each setpoint, and the uncertainty of methane concentrations determined by the RMSE of each of the applicable piecewise regression models. Most of the flux setpoints showed agreement between the measured flux quantities and the delivered. While some outliers are present, the results serve to validate the accuracy of the dynamic flux chamber for measuring across the range of 0–150 [g/m^2^-day]. The flux setpoints, measured quantities, and associated uncertainties are also documented in Table 8.

### 3.3. Field Validation Results

The flux chamber, equipped with calibrated sensors, were taken to the closed landfill to validate its performance in a field setting. On April 9th and 12th at least one hour of methane flux data was collected, with at least one hour of warmup time applied to the sensors. Figure 14 shows a time series comparison of measured methane concentration due to flux between the reference instrument (LI-COR) and the calibrated internal sensors. This data was collected on April 9th between 5:00 and 7:00 pm MST. While the trend observed using each measurement technique is similar, there was a noticeable offset between the two measurement techniques, resulting in an RMSE of 9.0 PPM and an MBE of 7.2 ppm. The sensors consistently overestimated the methane concentration compared to the LI-COR reference instrument. Another way to interpret this data can be seen in Figure 15, which shows the reference instrument measurements plotted against the sensor measured concentrations overlapping an ideal one-to-one line. The offset between the two measurement techniques is noticeable in the one-to-one plot too. After reviewing some leachate piezometer gas data on the landfill, we noticed that our sample site was near an area with relatively high hydrogen sulfide (H2S) (up to 500 ppm), and we believe the systematic error seen in Figure 14 and Figure 15 is likely attributable to relatively high concentrations of this confounding gas present.

For our next deployment we targeted an area of the landfill with no measurable H2S documented in nearby leachate piezometer wells. Figure 16 and Figure 17 show the time series and one-to-one plots, respectively, for the data collected on April 12th between 5:45 and 7:15 pm MST.

Table 9 and Table 10 summarize the RMSE and MBE of our internal methane and flux measurements, respectively, when comparing low-cost sensor measurements to reference measurements from the LI-COR gas analyzer.

The methane flux concentration data collected on April 12th shows much stronger agreement between the LI-COR and sensors than the field test on the 9th. The RMSE of 16.91 ppm is lower than the RMSE from testing the regression on the laboratory data in the 100–1500 ppm piecewise region, which was 21 ppm. We suspect this measurement location did not have a significant amount of confounding gases present in the methane flux. Discrepancies between our flux chamber sensor methane measurements and reference methane measurements during the first 15 min of the field test may be attributable to differential response times between our sensor arrays and the reference instrument. These results are promising but they highlight the importance of including a range of hydrogen sulfide concentrations and/or other confounding gases that could be present in landfill gas, like hydrogen, in future lab calibrations.

## 4. Conclusions

A low-cost, dynamic flux chamber was fabricated, calibrated, and validated for measurements of methane flux ranging from 0 to 150 g/m^2^-day. Using a centrifugal pc blower fan and a flow meter in series, the captured methane flux is dynamically passed through a chamber containing calibrated MOX sensors. The blower fan can maintain flow rates above 2.8 lpm; high enough to maintain a maximum methane concentration below the poisoning limit of the MOX sensors. The instrument has successfully operated autonomously numerous times with a power test displaying over 36 h of autonomous operation from a full charge while consistently transmitting all raw sensor data to a server for post-processing.

A methodology to calibrate the MOX sensors was developed by exposing the sensors to varying, controlled, steady state methane concentrations and wide ranges of temperature and humidity in a laboratory setting. The raw sensor data from the methane exposure was used to build a piecewise multiple linear regression spanning three unique methane concentration regions. The regions are 0–100 ppm, 100–1500 ppm, and 1500–12,000 ppm. The uncertainties (RMSE) for each piecewise region are 3.1 ppm, 21 ppm, and 307 ppm, respectively. We attempted to implement a 70-30 data split in the training of each regression, in which 30% of the data was used for validation during the training process. However, the limited size of our calibrations data made each individual data point highly unique, so using 100% of the data for training resulted in better model performance. In the future an autonomous calibration system could support larger training low-cost datasets with more redundancy and the application of a 70-30 split to enhance the model training process.

The calibrated sensors were used to measure laboratory-delivered flux concentrations which fed into a mass balance to quantify the predicted delivered methane flux. Transient and steady state models produced similar flux rates during a laboratory flux test, with the steady state model producing better agreement overall with the delivered flux rates. Since the delivered flux rates were held constant at each setpoint, this result is somewhat expected. A transient model may perform better in a setting with strong variation in flux rates with time.

After post-processing data from field deployments, we observed significant overestimation of methane (and flux) in a sampling location that was in relatively close proximity to a piezometer well with documentation of elevated concentration of H2S in the headspace. We suspect that the observed systematic error could be attributable to the influence of confounding gases emitted from the landfill in this area along with methane. Data from a field deployment further away from suspected confounding gases on the landfill yielded much better results. Some underprediction of methane was noted towards the beginning of the deployment which may be connected to our sensor array’s slower response time relative to the LI-COR optical gas analyzer.

To develop a regression which is effective against excess confounding gases, we recommend calibration data to be collected with various amounts of confounding gases, such as H2S and hydrogen (H2), present. H2S has been observed in some landfill gas from 0.2 to 2% [39]. H2 is often present in landfill gas up to 0.2% [40]. Given the relative abundance of these confounding gases and the highest expected concentration of CH4 in the flux chamber presented here at 150 g/m^2^/day, H2S and H2 setpoints delivered during a calibration would aim to span 0–125 ppm and 0–12.5 ppm, respectively. The ranges of temperature and humidity during any future calibrations should aim to cover the full parameter space anticipated in a given deployment location and time frame. A fully autonomous calibration system would make the vast number of calibration points required to improve measurement performance, especially with the addition of confounding gases, more tractable.

This study serves to establish the instrument’s viability to pursue a longer field validation for future work. Longer field deployments in the future that include periodic reference instrument co-locations with internal chamber CH4 measurements are needed to assess the regression models’ performance across more significant changes in environmental parameters over days, weeks, and seasons, as well as sensor drift. Future work should also assess the reproducibility in flux measurements across different flux chambers through laboratory validation and replicating field testing.

We had a goal of keeping the cost per unit below 1000 USD to make it feasible to construct and deploy multiple units across a landfill to help inform spatial variability in emissions. The per unit cost of this tool came in under our target budget at 730 USD. This breaks down into 130 USD for the data acquisition and transmission systems, 275 USD for solar power and storage systems, 200 USD for plumbing components, and 180 USD for the flux chamber and enclosures. This study demonstrates the feasibility of an operational low-cost dynamic flux chamber for landfill applications. The flux chamber and power system are readily portable by two people, allowing this tool to be readily deployed in rugged field conditions present in landfills.

## Figures and Tables

**Figure 1 sensors-25-06613-f001:**
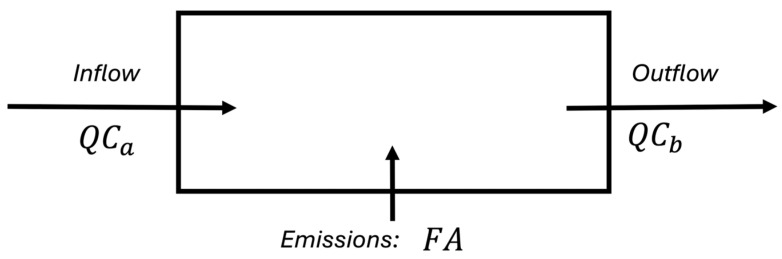
Mass balance diagram for a dynamic flux chamber.

**Figure 2 sensors-25-06613-f002:**
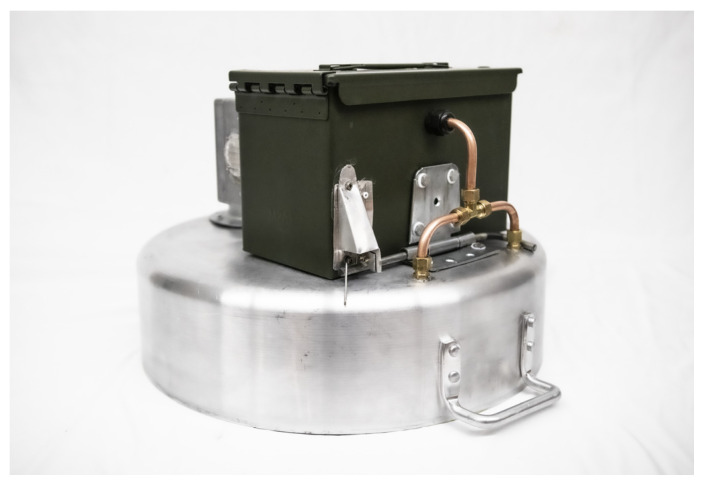
A photo of the dynamic flux chamber.

**Figure 3 sensors-25-06613-f003:**
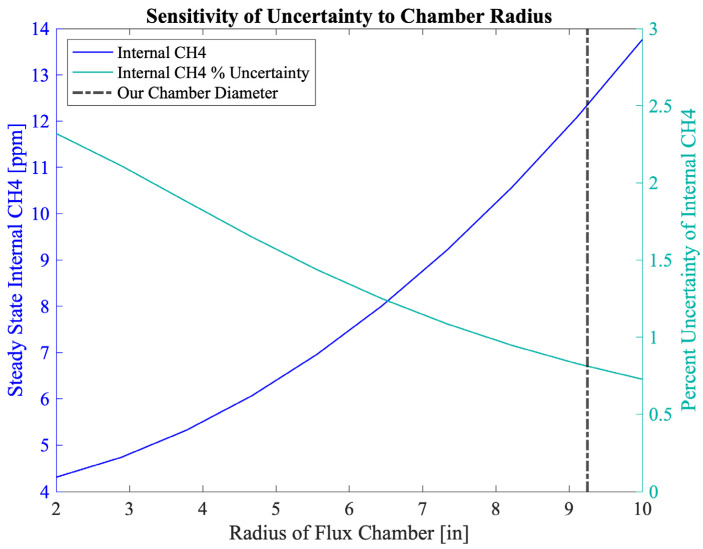
Sensitivity of internal chamber CH4 concentrations and their uncertainty at the low end of the flux measurement range to chamber diameter. Assumptions included: constant uncertainty of 10 ppm for internal CH4 concentrations, a flux rate of 0.1 g/m^2^/day, and a flow rate of 2.8 lpm.

**Figure 4 sensors-25-06613-f004:**
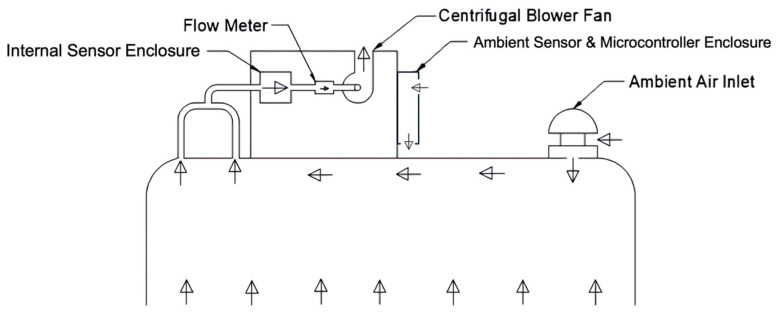
The flow path in the dynamic flux chamber. The arrows show the direction of the flow through the chamber.

**Figure 5 sensors-25-06613-f005:**
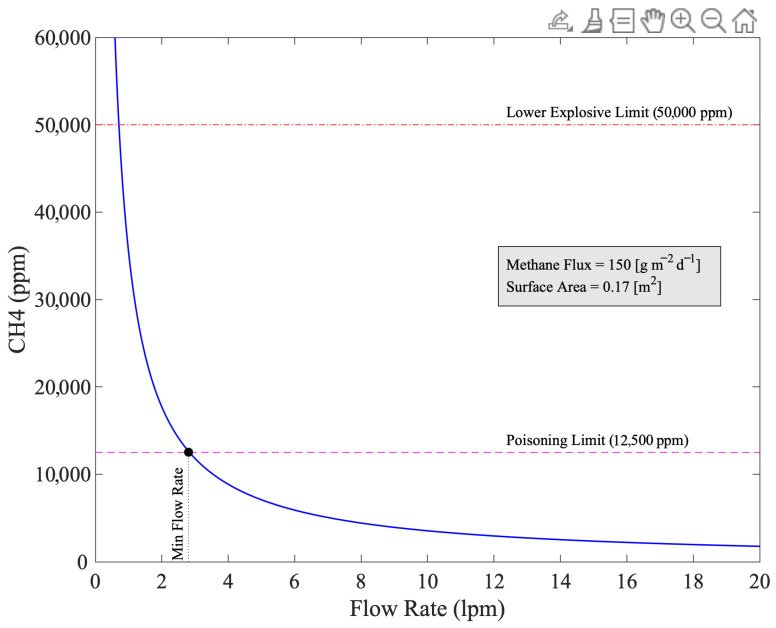
Expected steady state methane concentration in the chamber at different flow rates.

**Figure 6 sensors-25-06613-f006:**
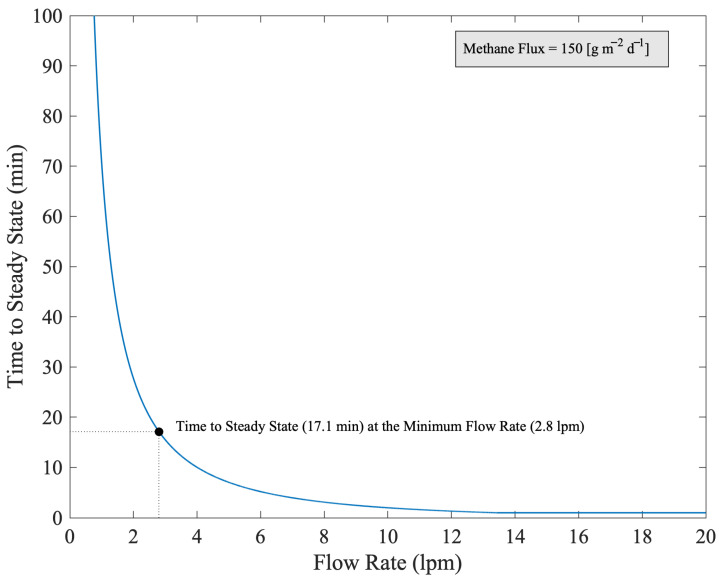
Expected time to steady state of methane concentration in the chamber. This is the longest anticipated time to steady state given the assumption of starting at an ambient concentration within the chamber and setting our flux to the maximum expected value of 150 g m^−2^ d^−1^.

**Figure 7 sensors-25-06613-f007:**
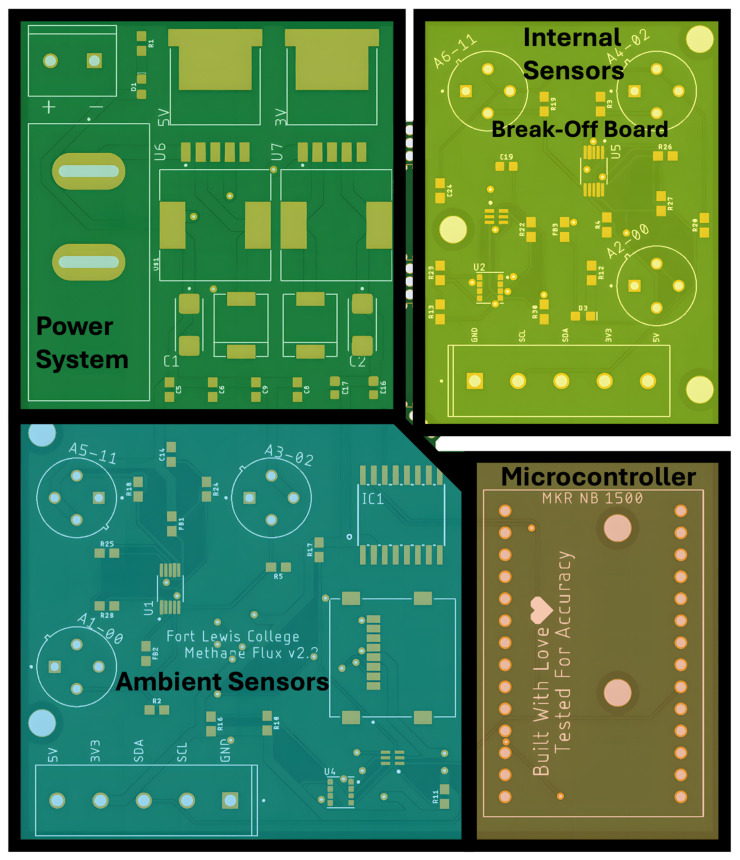
Diagram of the printed circuit board to show the general organization of components and the purpose of each section of the board.

**Figure 8 sensors-25-06613-f008:**
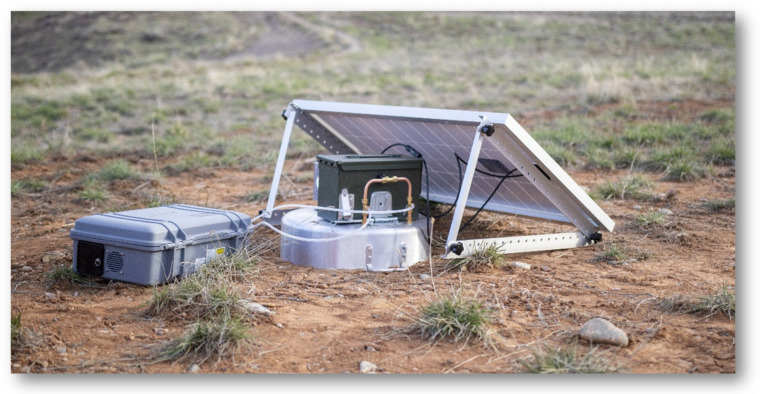
Flux chamber configuration for field validation.

**Figure 9 sensors-25-06613-f009:**
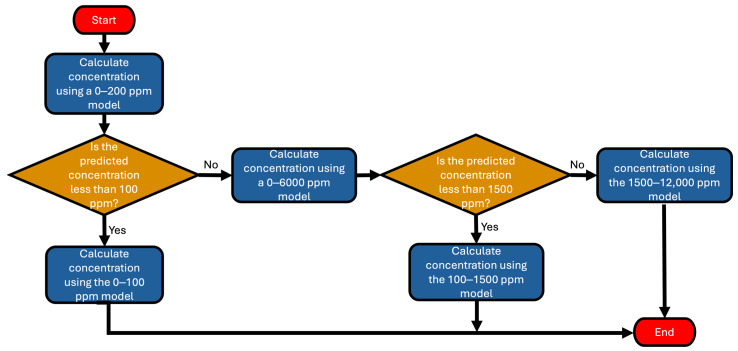
The flow chart above shows the process of choosing the applicable regression.

**Figure 10 sensors-25-06613-f010:**
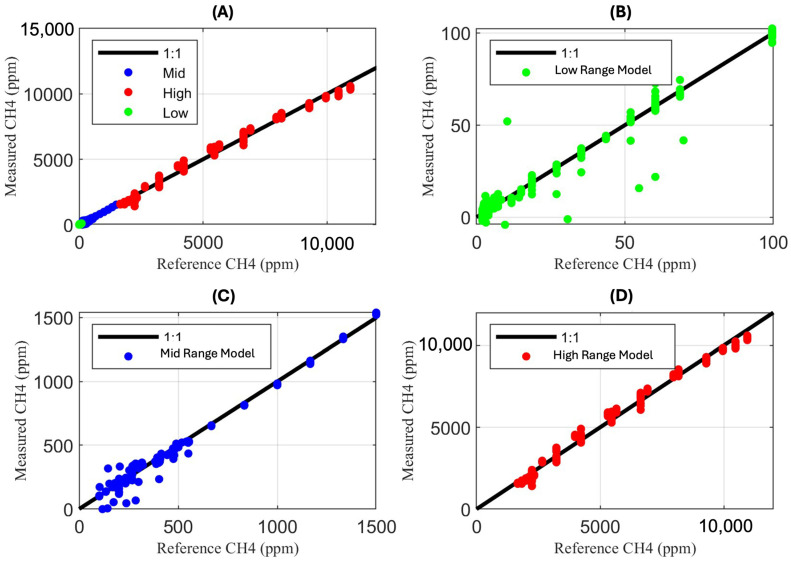
The plots above show the regression results using all laboratory data, including (**A**) the full range of measurements, (**B**) the low-range piecewise model for measurements between 0 and 100 ppm, (**C**) the mid-range piecewise model for measurements between 100 and 1500 ppm, and (**D**) the high-range piecewise model for measurements between 1500 and 12,000 ppm.

**Figure 11 sensors-25-06613-f011:**
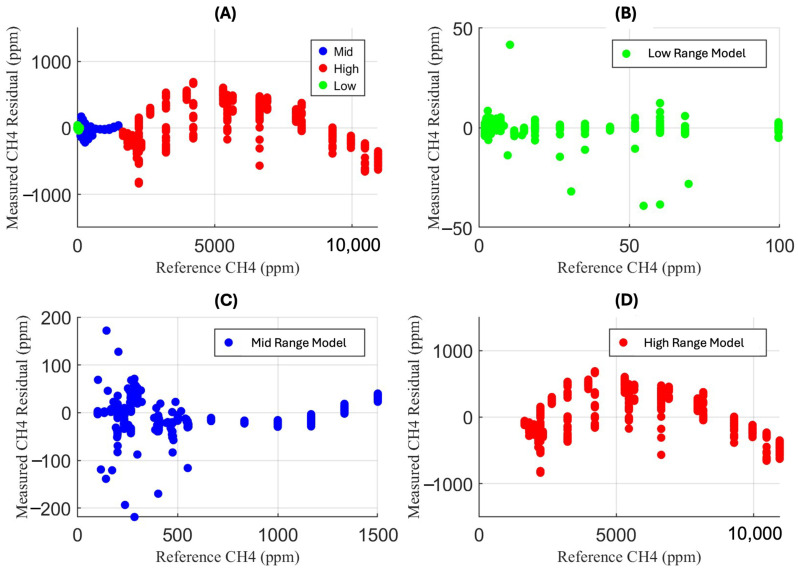
The plots above show the respective methane concentration residuals for each piecewise range for (**A**) the full range of measurements, (**B**) the low-range piecewise model for measurements between 0 and 100 ppm, (**C**) the mid-range piecewise model for measurements between 100 and 1500 ppm, and (**D**) the high-range piecewise model for measurements between 1500 and 12,000 ppm.

**Figure 12 sensors-25-06613-f012:**
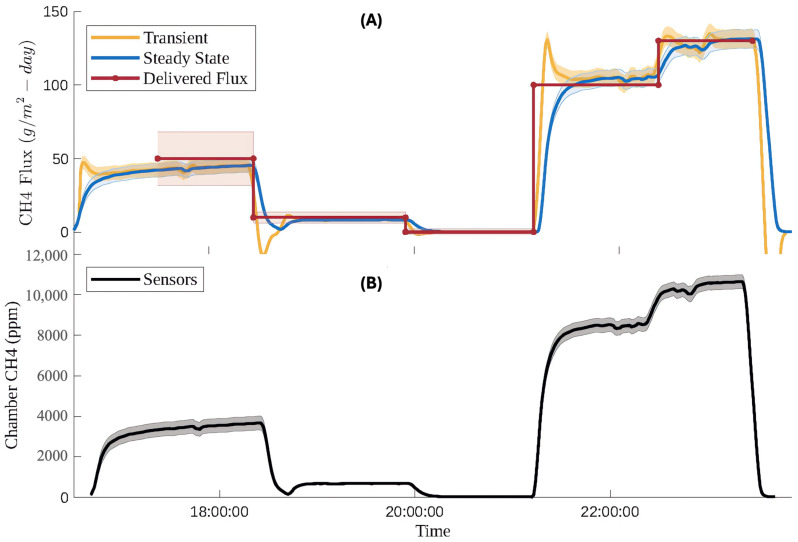
(**A**) Time series comparing delivered flux and measured flux when quantified with a continuous evaluation, a steady state model, and a continuous evaluation of a transient model. Shaded error bars represent the 95% confidence level of each parameter. For the delivered flux uncertainty is based on the uncertainty in the concentration of NIST traceable calibration gas and mass flow controller flows. For the steady state and transient calculated flux uncertainty is based on the manufacture specified uncertainty of the flow sensor, and the uncertainty of methane concentrations determined by the RMSE of each of the applicable piecewise regression models. (**B**) Time series of methane measurements from internal array of low-cost sensors. The shaded error bars represent the 95% confidence interval and are based on the RMSE of each of the applicable piecewise regression models.

**Figure 13 sensors-25-06613-f013:**
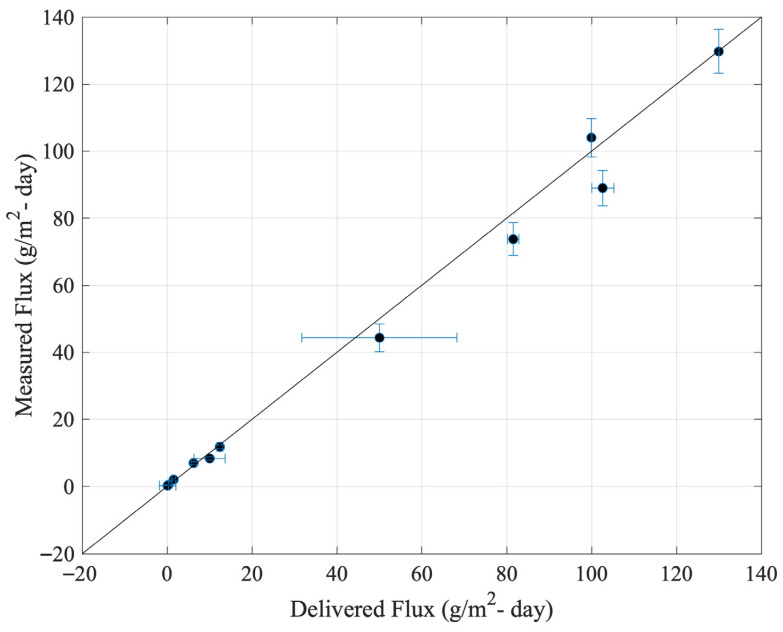
Measured steady state flux plotted against delivered flux with associated error bars and an overlapping one-to-one line. Flux rates at each setpoint calculated with the steady state model (using the average stabilized sensor response). Error bars represent a 95% confidence interval.

**Figure 14 sensors-25-06613-f014:**
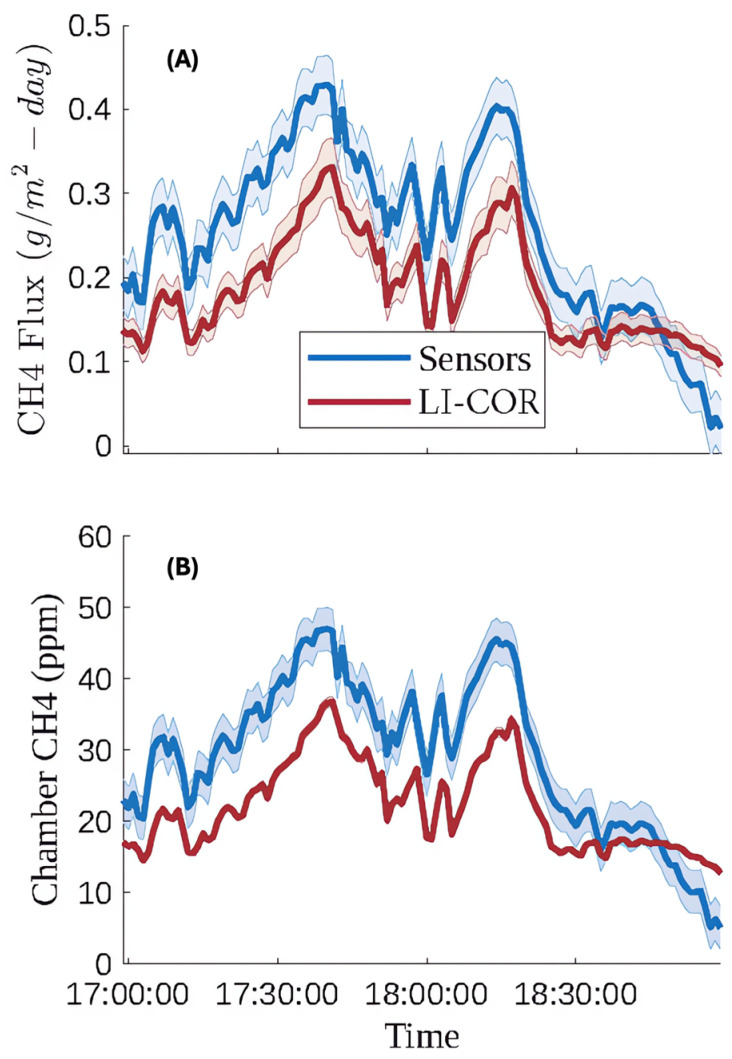
April 9th time series comparison between (**A**) the flux measurement determined by methane concentrations from the calibrated low-cost sensor array and the LI-COR reference instrument and (**B**) methane concentration measured by the LI-COR reference instrument and calibrated sensors. Shaded error bars for the LI-COR methane signal are defined by 2% uncertainty below 100 ppm and 10% uncertainty above 100 ppm, both of which were verified by delivering known gas concentrations. Shaded error bars for the methane concentrations measured by the internal low-cost sensor array represent a 95% confidence level and are based on the RMSE of each of the applicable piecewise regression models.

**Figure 15 sensors-25-06613-f015:**
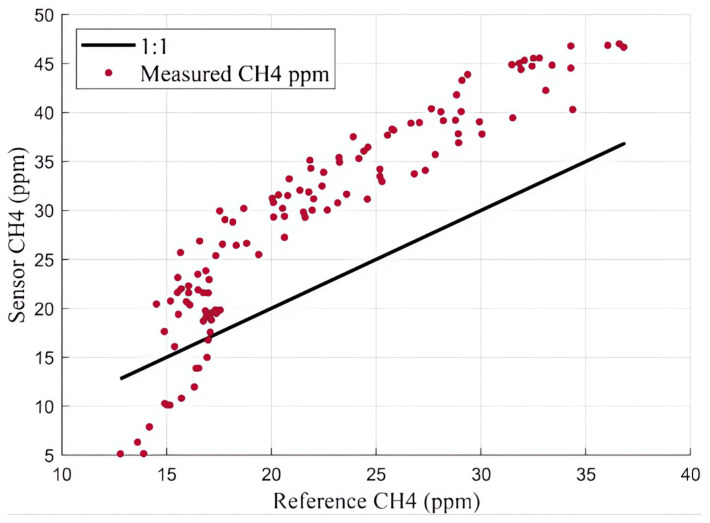
April 9th comparison between methane concentration measured by LI-COR reference instrument and calibrated sensors.

**Figure 16 sensors-25-06613-f016:**
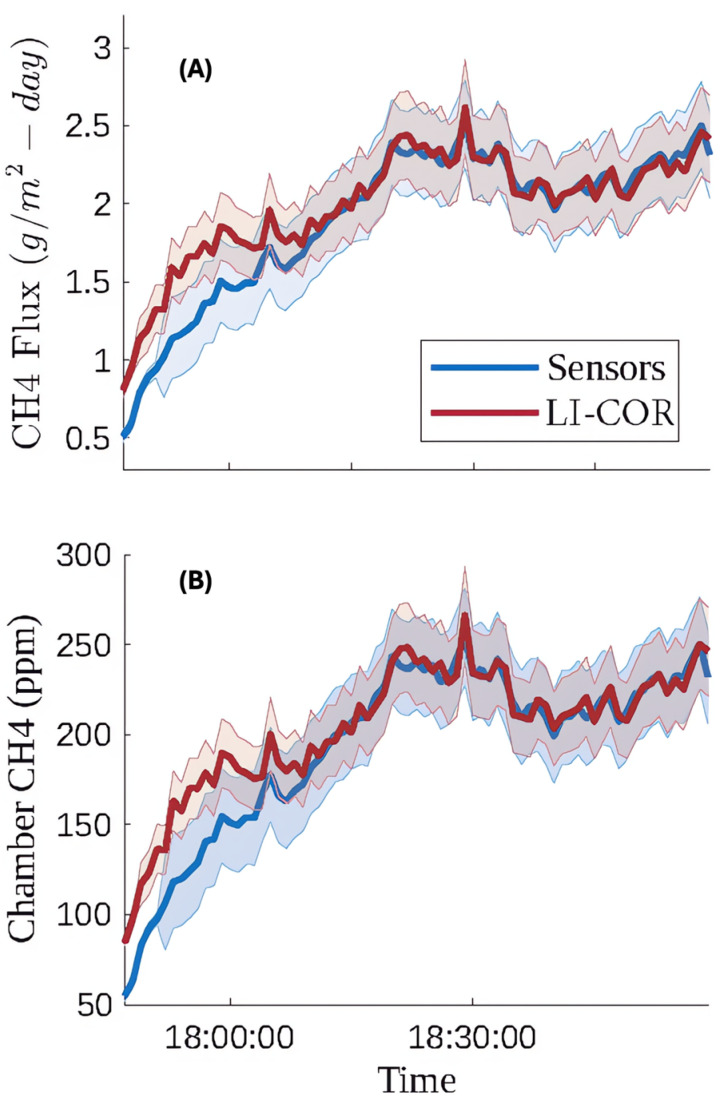
April 12th time series comparison between (**A**) the flux measurement determined by methane concentrations from the calibrated low-cost sensor array and the LI-COR reference instrument and (**B**) methane concentration measured by the LI-COR reference instrument and calibrated sensors. The RMSE of the measured CH4 relative to the reference is 16.91 ppm, but the percent error was much lower at 6.74%. Shaded error bars for the LI-COR methane signal are defined by 2% uncertainty below 100 ppm and 10% uncertainty above 100 ppm, both of which were verified by delivering known gas concentrations. Shaded error bars for the methane concentrations measured by the internal low-cost sensor array represent a 95% confidence level and are based on the RMSE of each of the applicable piecewise regression models.

**Figure 17 sensors-25-06613-f017:**
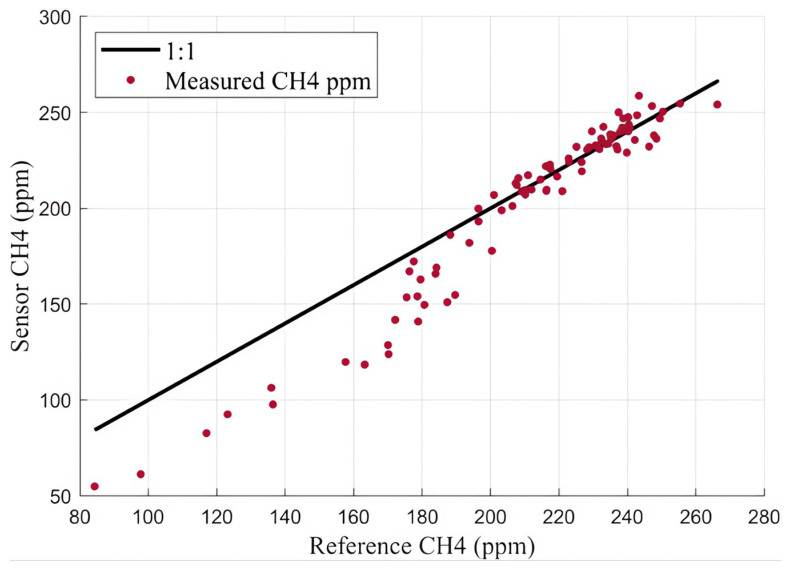
April 12th comparison between methane concentration measured by LI-COR reference instrument and calibrated sensors.

**Table 1 sensors-25-06613-t001:** Summary of tools and technology to measure CH4 emissions from landfills.

Technology/Tool	Strengths	Weaknesses
Aerial Remote Sensing [5]	Rapid, direct detection of methane plumes; identification of leaks and hotspots; relatively low-cost with use of UAVs	Limited spatial coverage; sensitive to weather and operator technique; typically qualitative; snapshot in time
Aerial Mass Balance [6,7]	Large area coverage; detection of major emission zones quickly; quantification of emissions of overall landfill area and hotspots in some cases	Expensive; snapshot in time; accuracy affected by wind and atmospheric conditions
Eddy Covariance [8,15]	Quantitative flux estimation; continuous measurements that can capture temporal variability; minimal site disturbance	Complex data analysis; requires specific meteorology; high-cost and technical expertise needed; relies on installation of tower infrastructure; limited spatial resolution of emissions
Fence-line Surveys with Gaussian Plume Modeling [9]	Tracer gas enhances accuracy; comprehensive spatial coverage and resolution possible	Model assumptions may not capture complex terrain or variable meteorology
Sensor Networks with Inverse Modeling [10]	Continuous monitoring over long periods; relatively low-cost	Indirect flux estimates with model uncertainty; requires dense, well-calibrated sensor network
Static Flux Chamber [11,13,17]	Simple, low-cost and direct measurement; portable	Labor-intensive; very limited spatial coverage; can alter local pressure and flux dynamics
Dynamic Flux Chamber [12,16,18]	Better accuracy relative to static chambers since concentrations build up in the chamber to a lesser extent; allows for continuous near-real-time flux determination; direct measurement	More complex systems relative to static chambers; limited spatial representativeness; requires stable setup and calibration; potential to alter flux dynamics related to pressure, temperature, and CH4 concentration
Targeted Satellite Observations [14,15]	Very broad spatial coverage; enables regional or global methane tracking; useful for identifying super-emitters	Limited spatial resolution; cloud interference; retrieval accuracy lower for small or variable sites

**Table 2 sensors-25-06613-t002:** Summary of sensor costs and sensitivities.

Maker	Sensor	Manufacturer Specified Sensitivities	Cost Per Unit	Ambient Sensor Array	Chamber Sensor Array
Figaro	TGS2600	air, methane, CO, iso-butane, ethanol, hydrogen, temperature, humidity	11.00 USD	1	1
Figaro	TGS2602	air, hydrogen, ammonia, ethanol, hydrogen sulfide, toluene, temperature, humidity	12.20 USD	1	1
Figaro	TGS2611	air, hydrogen, ethanol, methane, temperature, humidity	11.20 USD	1	1
Bosch Sensortec	BME680-Gas	ethane, isoprene, ethanol, acetone, CO, temperature, humidity	10.76 USD	1	1
	BME680-Pressure	pressure			
	BME680-Temperature	temperature			
	BME680-Humidity	relative humidity			

**Table 3 sensors-25-06613-t003:** Summary of 0–100 ppm piecewise regression.

Regression Range	Coefficient	Predictor	*p*-Value	Intercept
0–100 ppm	C_00_ = −0.00129	TGS2600	0.9634	B = −217.237
	C_02_ = 0.00501	TGS2602	3.013 × 10^−6^	
	C_11_ = 0.124537	TGS2611	7.872 × 10^−11^	
	C_BME_ = 1.443659	BME	3.921 × 10^−265^	
	C_T_ = −1.12251	Temperature	6.959 × 10^−21^	
	C_H_ = −0.15887	Relative Humidity	8.961 × 10^−5^	
	C_R1_ = −301.944	$R_{1}=\frac{BME}{TGS2611}$	0	
	C_R2_ = −77.486	$R_{2}=\frac{TGS2611}{TGS2602}$	3.249 × 10^−65^	
	C_R3_ = 66.68535	$R_{3}=\frac{TGS2600}{TGS2602}$	1.076 × 10^−18^	
	C_R4_ = 1581.711	$R_{4}=\frac{Temp}{TGS2611}$	1.948 × 10^−67^	
	C_R5_ = 72.63999	$R_{5}=\frac{TGS2611}{TGS2600}$	3.847 × 10^−10^	
	C_R6_ = −207.449	$R_{6}=\frac{BME}{TGS2602}$	3.514 × 10^−7^	

**Table 4 sensors-25-06613-t004:** Summary of 100–1500 ppm piecewise regression.

Regression Range	Coefficient	Predictor	*p*-Value	Intercept
100–1500 ppm	C_00_ = −2.01822	TGS2600	1.753 × 10^−255^	B = 768.135
	C_02_ = 0.106666	TGS2602	8.978 × 10^−109^	
	C_11_ = 1.159243	TGS2611	0	
	C_BME_ = 0.374516	BME	8.780 × 10^−14^	
	C_T_ = 12.47826	Temperature	1.220 × 10^−75^	
	C_H_ = 3.837942	Relative Humidity	3.673 × 10^−62^	
	C_R1_ = 361.3001	$R_{1}=\frac{TGS2600}{TGS2611}$	3.960 × 10^−46^	
	C_R2_ = −9359.08	$R_{2}=\frac{Temp}{TGS2611}$	1.185 × 10^−134^	
	C_R3_ = −765.343	$R_{3}=\frac{TGS2611}{TGS2600}$	0	
	C_R4_ = 0.467051	$R_{4}=\frac{TGS2611}{Humidity}$	2.519 × 10^−304^	
	C_R5_ = −0.09559	$R_{5}=\frac{TGS2602}{Humidity}$	2.356 × 10^−302^	
	C_R6_ = 902.1457	$R_{6}=\frac{BME}{TGS2602}$	2.880 × 10^−58^	

**Table 5 sensors-25-06613-t005:** Summary of 1500–12,000 ppm piecewise regression.

Regression Range	Coefficient	Predictor	*p*-Value	Intercept
1500–12,000 ppm	*C*_00_ = −2.84299	TGS2600	6.607 × 10^−169^	B = 2952.922
	*C*_02_ = 0.033676	TGS2602	0.0825	
	*C*_11_ = 4.499467	TGS2611	0	
	*C_BME_* = −1.11855	BME	3.927 × 10^−10^	
	*C_T_* = −189.655	Temperature	0	
	*C_hum_* = −180.961	Relative Humidity	0	
	C_R1_ = −434,102	$R_{1}=\frac{Temp}{TGS2611}$	0	
	C_R2_ = −39.383	$R_{2}=\frac{TGS2611}{BME}$	0	
	C_R3_ = −3907.23	$R_{3}=\frac{TGS2611}{TGS2600}$	0	
	C_R4_ = 387,837.2	$R_{4}=\frac{Temp}{TGS2600}$	0	
	C_R5_ = 293,553.9	$R_{5}=\frac{Humidity}{TGS2611}$	1.284 × 10^−246^	
	C_R6_ = 5562.674	$R_{6}=\frac{Humidity}{BME}$	7.929 × 10^−184^	

**Table 6 sensors-25-06613-t006:** Summary of the regression performance for each piecewise range.

Piecewise Range	RMSE (ppm)	MBE	R^2	Max Percent Error (%)
10–100	3.1	2.9 × 10^−14^	0.994	31.0
100–1500	21	−0.38	0.997	21.1
1500–12,000	307	1.1	0.991	20.5

**Table 7 sensors-25-06613-t007:** Summary performance during lab flux test, comparing transient and steady state flux quantification methods.

Quantification Method	RMSE (g/m^2^-Day)	MBE (g/m^2^-Day)
Transient Model (Continuous)	50	−4.9
Steady State Model (Continuous)	10.8	−1.6
Setpoint Average (Discrete)	7.3	−2.2

**Table 8 sensors-25-06613-t008:** Summary delivered and measured flux rates during the laboratory validation test.

Delivered Flux (g/m^2^-Day)	Measured Flux (g/m^2^-Day)	% Uncertainty of Measured Flux
0.11 ± 2	0.26 ± 0.04	15
0.33 ± 0.06	0.44 ± 0.04	10
1.6 ± 0.5	2.03 ± 0.3	13
6.3 ± 0.5	6.9 ± 0.4	6
10.0 ± 4	8.3 ± 0.4	5
12.5 ± 0.7	11.8 ± 0.6	5
50 ± 20	44 ± 4	9
82 ± 1	74 ± 5	7
100.0 ± 0.1	104 ± 6	5
103 ± 3	89 ± 6	6
130.0 ± 0.2	129 ± 7	5

**Table 9 sensors-25-06613-t009:** Summary of the field-testing chamber methane measurement performance. These metrics represent our low-cost sensor CH4 measurement’s deviation from our reference instrument’s CH4 measurement, without accounting for the propagated error of each measurement.

Test Date	RMSE (ppm)	MBE (ppm)	Mean % Error
April 9th	9.0	7.2	37.0
April 12th	16.9	−7.6	6.74

**Table 10 sensors-25-06613-t010:** Summary of the field-testing flux measurement performance. These metrics represent the deviation of the flux rates calculated with chamber CH4 measurements from low-cost sensors, from the flux rates calculated with CH4 measurement made with the reference instrument, without accounting for the propagated error of each measurement.

Test Date	RMSE (g/m^2^-Day)	MBE (g/m^2^-Day)	Mean % Error
April 9th	0.08	0.05	30.1
April 12th	0.47	−0.31	11.5

## Data Availability

The original data presented in the study are openly available at https://github.com/justone84/CH4-Flux-Instrument (accessed on 1 October 2025).

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
