# Peer review of "Calibration and Validation of an Autonomous, Novel, Low-Cost, Dynamic Flux Chamber for Measuring Landfill Methane Emissions"

_sensors, 2025, doi:10.3390/s25216613_

Round 1
Reviewer 1 Report
Comments and Suggestions for Authors
This paper presents the design, calibration, and validation of a low-cost autonomous dynamic flux chamber that uses metal oxide (MOX) sensor arrays and piecewise multiple linear regression to measure methane emissions from landfills. Laboratory and field tests show that the chamber can reliably quantify methane fluxes (0–150 g/m²-day), with accuracy limited mainly at very low concentrations and in the presence of interfering gases such as H₂S. Here are some comments for authors that may help to improve their manuscript:
1- A sensitivity analysis on how chamber size and volume affect flux accuracy would strengthen the justification.
2- Since pressure imbalances can strongly bias flux estimates, more discussion is needed on how chamber pressure was monitored or stabilized during operation.
3- The study applied multiple linear regression (MLR). Did the authors explore or compare nonlinear models or machine learning methods, which might reduce residuals at low concentrations?
4- In the literature review of low-cost MOX sensors, refer to the new study of room temperature gas sensors, such as DOI: 10.1016/j.snb.2022.131418.
5- The paper uses a piecewise calibration strategy. Could the authors justify the chosen breakpoints (100 ppm and 1500 ppm) and whether they are based on sensor behavior or landfill concentration distributions?
6- How robust is the calibration over time? Is there evidence of sensor drift, and if so, how often should recalibration be performed in real deployments?
7- Were replicate chambers or repeated deployments tested to evaluate reproducibility and robustness across units?
8- The conclusions suggest expanding calibration datasets with confounding gases. Could the authors propose a structured plan for this (e.g., lab matrix experiments including H₂S, CO₂, humidity extremes)?
Reviewer 2 Report
Comments and Suggestions for Authors
Please see the attached file

Reviewer 3 Report
Comments and Suggestions for Authors
The authors presented an interesting article on the dynamic flux chamber for measuring landfill methane emissions, which is a pressing problem of modern times. The absolute advantage of this article is the description of the first dynamic flux chamber for measuring methane emissions using low-cost sensors. The authors conducted an in-depth study of the influence of external factors (methane concentration, temperature and humidity) on the building of a multiple linear regression model. The article is well organized but needs a bit of tweaking before publication in my opinion.
The field study used a very small number of measurements. The dynamic flux chamber operated for only a couple of hours, of which only about an hour was spent collecting data. Therefore, the collected data lacked information about changes in environmental parameters over days, weeks, and seasons. It is not clear how correctly the regression presented by the authors would work over longer time periods.
The text lacks a more detailed description of the choice of coefficients in the regression. Why were these particular values chosen for the C coefficients? How significantly did the coefficients differ between the laboratory and field studies? Was a Monte Carlo analysis completed to iteratively test unique relationships to optimize the RMSE based on field-obtained data? Did the optimal predictor ratios differ between the laboratory and field studies?
Author Response
Please see the attachement.
